# The Role of Ketogenic Diet in the Treatment of Neurological Diseases

**DOI:** 10.3390/nu14235003

**Published:** 2022-11-24

**Authors:** Damian Dyńka, Katarzyna Kowalcze, Agnieszka Paziewska

**Affiliations:** 1Institute of Health Sciences, Faculty of Medical and Health Sciences, Siedlce University of Natural Sciences and Humanities, 08-110 Siedlce, Poland; 2Department of Neuroendocrinology, Centre of Postgraduate Medical Education, 01-813 Warsaw, Poland

**Keywords:** ketogenic diet, neurological diseases, epilepsy, Alzheimer’s disease (AD), Parkinson’s disease (PD), multiple sclerosis (MS), migraine, brain, neurone, neuroinflammation, ketone bodies, ketogenic, neurotransmitters, neuroplasticity, treatment, prevention, inflammatory, anti-inflammatory, low carb, high fat, nutrition

## Abstract

Over a hundred years of study on the favourable effect of ketogenic diets in the treatment of epilepsy have contributed to a long-lasting discussion on its potential influence on other neurological diseases. A significant increase in the number of scientific studies in that field has been currently observed. The aim of this paper is a widespread, thorough analysis of the available scientific evidence in respect of the role of the ketogenic diet in the therapy of neurological diseases such as: epilepsy, Alzheimer’s disease (AD), Parkinson’s disease (PD), multiple sclerosis (MS) and migraine. A wide range of the mechanisms of action of the ketogenic diet has been demonstrated in neurological diseases, including, among other effects, its influence on the reduction in inflammatory conditions and the amount of reactive oxygen species (ROS), the restoration of the myelin sheath of the neurons, the formation and regeneration of mitochondria, neuronal metabolism, the provision of an alternative source of energy for neurons (ketone bodies), the reduction in glucose and insulin concentrations, the reduction in amyloid plaques, the induction of autophagy, the alleviation of microglia activation, the reduction in excessive neuronal activation, the modulation of intestinal microbiota, the expression of genes, dopamine production and the increase in glutamine conversion into GABA. The studies discussed (including randomised controlled studies), conducted in neurological patients, have stressed the effectiveness of the ketogenic diet in the treatment of epilepsy and have demonstrated its promising therapeutic potential in Alzheimer’s disease (AD), Parkinson’s disease (PD), multiple sclerosis (MS) and migraine. A frequent advantage of the diet was demonstrated over non-ketogenic diets (in the control groups) in the therapy of neurological diseases, with simultaneous safety and feasibility when conducting the nutritional model.

## 1. Introduction

The extreme and dynamic increase in incidences of neurological diseases (including neurodegenerative ones) constitutes a real problem for many millions of people worldwide. This phenomenon is all the more so alarming when taking into account the results of the Global Burden of Disease (GBD) study. It has been shown, indeed, that widely understood neurological diseases are the second most frequent cause of death worldwide [1]. Moreover, they are the most frequent cause of disability and, therefore, an increase in the DALY (disability-adjusted life years) index, i.e., years of life lost due to health damage or preterm death. For comparison, attention is called to the fact that the problem is far greater than the declared COVID-19 pandemic, and for that reason, it is worthwhile to take a more detailed look at this issue [2]. It seems justified to pose the question of whether this can be associated with an even greater epidemic of neurological diseases in the coming years. Although possible post-COVID-19 complications in the form of neurological disorders (including neurodegenerative ones) have been reported [3,4,5], it has been, however, demonstrated that a positive COVID-19 test result causes no greater risk of neurological disease occurrence than other respiratory tract infections. One exception is ischaemic stroke [6], but the possibility of a twice higher risk of epilepsy development has also been suggested [7]. Taking into account the increasing incidence of neurological diseases, a need is seen for a complex approach, looking for new solutions or an analysis of the already existing ones in order to refine or draw new conclusions based on the most recent data. One of the ways of such a complex therapeutic approach to neurological diseases is the nutritional aspect. Taking into account over a hundred years of study and clinical practice, the ketogenic diet is worth particular attention in this respect.

The ketogenic diet is a method of nutrition leading to the increased production of ketone bodies (β-hydroxybutyrate, acetoacetate and acetone) in the organism and, thus, to a condition of ketosis. This effect comes about by obtaining the greatest energy share from fats and minimising the consumption of carbohydrates. Importantly, a ketosis condition can also be obtained, among other methods, through fasting or the observation of a very low-calorie diet (not necessarily with a predominance of fats); however, that condition will not be nutritional ketosis [8,9]. Keeping in mind this minor reservation, it is difficult to refine a universal definition of the ketogenic diet, although the most accurate definition seems to be the one mentioned above. Transforming the theoretical considerations into practice, the diet can be expressed as a share of the macronutrients within the following range: fats 60–90% (usually 70–75%) of the whole energy content of the diet, carbohydrates below 50 g daily (which usually accounts for 5–10% of the whole energy content of the diet) and protein 1.0–1.2 to 1.7 g per kg body weight (which usually accounts for about 20% of the whole energy value of the diet) [10,11,12,13,14,15]. The ketogenic diet, as a rule, should imitate a fasting condition in the body, leading, however, to no negative effects of starvation. Both fasting and the observation of the ketogenic diet lead to increased oxidation of fatty acids and the use of the ketone bodies produced in that process (β-hydroxybutyrate, acetoacetate and acetone) as the main energy substrate. Therefore, the diet distinguishes itself from other diets since it is not based on energy obtained from glucose but on that from ketone bodies produced from fats [8,16,17,18,19,20].

The property of mimicking a fasting condition is of particular importance in the context of the subject of this paper. This is associated with the fact that even before the “creation” of the ketogenic diet, a fast was proposed in order to help neurology patients with epilepsy, after which an evident improvement was observed. That practice has a very long tradition in history since it can be found that many sources, even those from the 5th century before Christ, have described the favourable effects of fasting in cases of epilepsy. They suggested that fasting is an effective method of treatment for epileptic seizures [21]. Other historical data mention that Hippocrates suggested a need for the limitation of calories in the treatment of epilepsy [22]. Some mentions of the topic can also be found in biblical texts. The Gospel according to Saint Mark describes how Jesus cured a boy of epilepsy, saying that it could be treated with fast and prayer. Interestingly, the whole scene of the cure is shown in the painting “The Transfiguration” by Raphael [23,24,25]. In the less distant past, i.e., in 1911, doctors from Paris reported the benefits resulting from the observation of fasting during the treatment of epilepsy based on their own medical practice [26]. The 20th century was the time at which fasting and epilepsy were increasingly and frequently the subjects of scientific research and not only clinical practice [22]. The real breakthrough was the moment at which it was discovered that a ketogenic diet could show fast-like therapeutic properties in epilepsy but without the negative consequences of starvation (i.e., malnutrition). The paper by Wilder in 1921, which was the beginning of ketogenic diet use in the treatment of epilepsy, can serve as a point of reference [22,27,28]. The proportions of macronutrients in a clinical ketogenic diet were then proposed as 10–15 g of carbohydrates daily and 1 g of protein per kg body weight; the rest of the energy share should be provided by fats [29]. The extremely favourable results observed meant that from that time on, the use of the ketogenic diet in epilepsy became increasingly widespread, to such an extent that it was recommended in textbooks as the standard therapy of that disease [25]. Over the years, the diet has been increasingly extensively studied in various diseases; however, its main range of use for over a hundred years has included neurological diseases (initially used in epilepsy). Taking into account its primary use and the many years of practice, it seems justified to study its potential application in other neurological diseases (including neurodegenerative ones). The perspective of over a hundred years since its breakthrough application means that a meticulous analysis of current knowledge concerning not only epilepsy but also other neurological diseases seems to be of great importance for the development of science in this respect. The intensification of studies on the effect of the ketogenic diet on neurological diseases is well reflected in Figure 1 showing the number of publications in the PubMed search engine after entering the words “ketogenic diet neurological disease”. A significant increase in the number of publications can be seen at the beginning of the 21st century, with a significantly growing tendency in the subsequent years.

## 2. The Range and Mechanisms of the Ketogenic Diet Effect in Neurological Diseases

The potential mechanisms of the ketogenic diet are multifaceted. The extremely wide range of its effect on neurological diseases is described in a publication of 2021, in which the authors thoroughly analysed 170 studies and described the range of its effect in 14 various aspects, e.g., on neuroprotection, neuroplasticity, neuroinflammation, function of neurotransmitters, epigenetics, nociception, changes in cell energetics and metabolism, and other aspects [30]. An important piece of information is the fact that the brain can function excellently, taking energy from ketone bodies as the energy substrate [31]. Frequently, that is the only option for energy provision when glucose is not readily available [32]. The cerebral cells, in fact, contain monocarboxylate transporters (MCTs), through which ketone bodies are transported in order to provide energy, similar to many other body cells [33]. The wide and, in the first place, favourable range of the ketogenic diet effect was presented in a widespread meta-analysis in 2021, based on 49 animal studies from the years 1979–2020. Strong neuronal protection was demonstrated against acute central nervous system damage as a result of a reduction in the death rate, damage and dysfunction of neurons [34]. Moreover, the frequently used calorie deficit, together with a ketogenic diet, also shows an additional neuroprotective potential. It increases, in fact, the number of neuroprotective factors, i.e., the brain-derived neurotrophic factor (BDNF) and the glial cell line-derived neurotrophic factor (GDNF), neutrophin-3 (NT-3), and molecular chaperones. The deficit also improves mitochondrial function (and thus increases the efficiency of energy production and reduces reactive oxygen species (ROS) production). It also shows an anti-inflammatory potential in the result of, among other effects, the inhibition of the activities of cyclooxygenase-2 (COX-2) and inducible nitric oxide synthase (iNOS) and in the result of a blockade of the synthesis of proinflammatory interleukins (IL-1β, IL-2, IL-4, IL-6) and tumour necrosis factor alpha (TNFα). It also reduces the level of the central component of the inflammatory process, i.e., transcriptional nuclear factor kappaB (NFκB) [35,36].

Many neurological diseases (including neurodegenerative ones) are characterised by glucose metabolism disturbances in the neurons. Although not the only one, the most characteristic example is traumatic cerebral damage, which includes brain energetic collapse due to major mechanical trauma. The group at particular risk are sportsmen practising contact sports (i.e., martial arts, American football). They are thus at risk of repeated brain concussion episodes, which soon lead to chronic post-traumatic encephalopathy. Extremely important is the fact that in the situation of brain trauma and energetic collapse, the number of MCT channels (which transport ketone bodies to cells) in the brain cells increases by 85%; the number of β-hydroxybutyrate-metabolising enzymes also increases [37,38,39,40,41,42]. Based on that, we can say that the brain demands another source of energy, which can be provided by ketone bodies. Most recent studies also show that in patients with traumatic cerebral damage, in whom a ketogenic diet was applied, no clinical adverse effects were noted, and its safety and applicability were found [43]. The application of the ketogenic diet, through the mediation of β-hydroxybutyrate, can reduce demyelination, the death of oligodendrocytes (producing myelin) and the degeneration of axons caused by glucose deficiency [44]. The widely understood damage can also concern the mitochondrial structures at the level of respiratory chain complexes. It has been demonstrated that β-hydroxybutyrate, on one hand, provides components participating in the reconstruction of the respiratory chain; on the other, energy acquisition from ketone bodies is possible even in the case of damage to the first complex of the respiratory chain [45,46]. For brain functioning, even in healthy individuals, an increase in β-hydroxybutyrate concentration can be far more favourable in comparison to glucose. This has been confirmed in a study in which hydroxybutyrate infusion in healthy individuals (until 5.5 mmol/L concentrations are obtained), compared with the lack of such infusions, reduced cerebral glucose utilisation by 14% and caused an increase of cerebral blood flow by as much as 30%, with unchanged oxygen consumption. The authors directly suggest a possibility of the neuroprotective effect of ketone bodies [47].

In many neurodegenerative diseases, the problems include the development of amyloid plaques, which, in turn, are strongly correlated with high glucose concentration, diabetes mellitus and insulin resistance [48,49]. Such problems most frequently accompany high-carbohydrate diets [50]. Taking into account the strongly hypoglycaemic effect of a ketogenic diet and the evident insulin-concentration-reducing effect [51,52], it seems reasonable that a ketogenic diet would have a favourable effect on the prevention of the deposition of amyloid plaques or a reduction in their number. It has been found that studies have actually demonstrated such an effect of the diet, which, however, goes beyond the effect on glycaemia and may be associated, i.a., with autophagy as well. The condition of ketosis can promote macroautophagy in the brain through the activation of sirtuin 1 (SIRT1) and hypoxia-induced factor 1α (HIF-1α) and the inhibition of the mTORC1 complex. This can prevent neurodegenerative disorders, among other effects, through the elimination of protein aggregates or damaged mitochondria [53,54]. It thus prevents the accumulation of autophagosomes and improves the survival of cortical neurons, leading to a reduction in cerebral damage [55,56,57]. Another study has also revealed that ketogenic diets can effectively reduce the amount of beta-amyloid and also other redundant by-products of metabolism in cerebral tissue [58].

Among other potential mechanisms of influence of the ketogenic diet on the nervous system and neurological diseases, its anti-inflammatory activity can be mentioned, which is multifaceted [59]. A neuronal inflammatory condition can result, i.a., from trauma, ischaemia, degeneration or infection [60]. The ketogenic diet can exert effects on the regulation of both central and peripheral inflammatory mechanisms [61]. Its effect has been shown in a reduction in microglia activation and the decreased expression of proinflammatory cytokines, i.a., IL-1β, IL-6 and TNF-α in the hippocampus. Moreover, it can inhibit neuritis through the suppression of the cyclooxygenase-2 (COX-2)-dependent pathway by peroxisome proliferator-activated receptor γ (PPARγ) activation [62,63,64]. Moreover, astrocytes (glial cells) present in the brain have the properties of ketone body production, and for that reason, it is suggested that this fact can exert a neuroprotective effect [65]. The possible (only anti-inflammatory) mechanisms of ketogenic diet action are much more numerous [66,67]; therefore, their anti-inflammatory potential possibly results from their synergy.

We also discuss the possible effect of a ketogenic diet on the blood–brain barrier (BBB). Although no unequivocal evidence is currently available, based on current knowledge, it is justified to look at that topic from a future-oriented perspective. Similar to the intestinal barrier, here, a break of membrane integrity can also occur. This happens, among other conditions, in epilepsy, Alzheimer’s disease (AD) and other neurological diseases [68,69,70]. In the period of an increased concentration of ketones in the blood, the permeability of the blood–brain barrier for β-hydroxybutyrate also increases [71]. The condition of ketosis can cause the restoration of the integrity of the blood–brain barrier as a result, i.a., of an increased content of connexin-43 (Cx43) in building the barrier and of monocarboxylate transporters (MCTs) and (glucose transporter) GLUT transporters [8,72,73]. Additionally, the ketogenic diet promotes the outflow of the above-mentioned amyloid plaques across the blood–brain barrier because it increases the concentration of proteins participating in the elimination of amyloid plaques, i.e., LDL receptor-related protein 1 (LRP1), glycoprotein P (P-gp) and phosphatidylinositol binding clathrin assembly protein (PICALM) [74].

Another possible mechanism of ketogenic diet activity in the nervous system includes, among other effects, an ability to change the cerebral metabolism of glutamine. As Yudkoff et al. believe, the condition of ketosis can intensify the metabolism of astrocytes, which results in the increased conversion of glutamate into glutamine. This can contribute to a reduction in the main stimulatory neurotransmitter (glutamate) concentration, with a simultaneous increase of the main inhibitory neurotransmitter (GABA) level, resulting in the intensified conversion of glutamine into GABA [75,76]. Thus, a decrease in excitotoxicity and greater mood improvement occur as the result of physiological nervous quietening. On the other hand, excitotoxicity inhibition, increased resistance to stress and an effect on synaptic plasticity can also result from the β-hydroxybutyrate effect increasing mitochondrial respiration, leading to a change in brain-derived neurotrophic factor (BDNF) expression [77]. That is one of the reasons why in individuals on ketogenic diets, frequently, a greater quietening, better concentration, mood improvement and an increase in cognitive abilities are observed [78,79].

The remaining possible favourable mechanisms of the ketogenic diet’s influence on the nervous system include, i.a., the effect on mitochondria (particularly important in neurological diseases), which has been increasingly widely discussed in the latest publications [80]. Apart from the earlier described indirect effect on mitochondria, through action at the gene level, the diet shows an ability to promote the biogenesis of mitochondria in the hippocampus [81]. Another mechanism, increasingly frequently mentioned in publications, is the effect of the ketogenic diet on the prevention and treatment of neurological diseases (including neurodegenerative ones) through a change of intestinal microbiota [82]. This results from a possible influence of intestinal dysbiosis on the development of such diseases, while on the other hand, it is known that a ketogenic diet significantly affects the remodelling of the intestinal microbiota [83]. Thus, increasingly, the subject of the potential indirect favourable effect of the ketogenic diet on neurological diseases through the modulation of intestinal microbiota is discussed. This is all the more justified when taking into account the direct connection of the intestine with the brain by the gut–brain axis, through which microbiota can affect brain processes (that concern, i.a., neurotransmission) and vice versa [84]. The ketogenic diet can also affect the processes of neurogenesis, that is, brain regeneration, the development of new nervous cells and their linking in neuronal networks [85,86]. Potential mechanism of ketogenic diet effect in neurological diseases are illustrated in Figure 2. Keeping in mind all the mentioned processes and the remaining mechanisms of the possible effect of the ketogenic diet on the prophylaxis and treatment of neurological diseases (including neurodegenerative ones), the arguments suggesting an unquestionable need for further science development in this field and an extension of the knowledge in this respect seem to be well-grounded. This would allow us to discover the existing (including those not yet known) neuroprotective mechanisms of the ketogenic diet.

## 3. The Role of the Ketogenic Diet in the Treatment of Epilepsy

Epilepsy is a neurological disorder associated with constant recurrent seizure attacks. The disease itself is not fraught with a great risk of death, and in most patients, the prognosis (measured based on the absence of seizures) is favourable. It is, however, associated with a significant impairment in the quality of life. Although it concerns all age groups (and either sex), it is one of the most frequent neurological disorders of childhood (it occurs slightly more frequently in males). About 50 million people struggle with the disease worldwide. The mean incidence of active epilepsy is 6.38 cases per 1000 population. It affects, to a greater extent, the populations of countries with low and medium incomes. Hence, millions of people worldwide are affected. It is worth stressing, however, that not all individuals with seizures have epilepsy since seizures can also occur, e.g., due to acute injury to the central nervous system [87,88,89,90]. Currently, as a rule, epileptic patients are prescribed drugs, which, however, give no benefit to some patients diagnosed with so-called drug-resistant epilepsy. This concerns about one-third of all epileptic patients, including 7–20% of children and 30–40% of adults [91,92]. In such cases, a ketogenic diet may be the only rescue, which has been confirmed, among other studies, by an extensive meta-analysis in 2020 [93].

Although a hundred years ago, the ketogenic diet was the main therapeutic option for epilepsy, after 1940, pharmacological treatments became the main therapeutic method. An antiepileptic drug called Dilantin superseded the use of the ketogenic diet to a significant extent. Its effectiveness in reducing epileptic seizures caused doctors to be more willing to reach for that method of therapy, less and less frequently ordering ketogenic diets [25]. That resulted, of course, from obtaining similar results with a lower amount of work (since the institution of the diet requires a significantly wider medical approach than prescribing a drug). As mentioned above, pharmacotherapy has, however, failed and still fails in about 1/3 of cases. For that reason, the ketogenic diet, after over a hundred years, still is one of the standard therapeutic options in the case of drug-resistant epilepsy when pharmacotherapy fails. A renaissance of the ketogenic diet was observed in 1997 after the movie “First do no harm”, presenting the experience of Charlie Abrahams (son of Jim Abrahams, a Hollywood film producer) with the ketogenic diet, which proved to be the last resort for epileptic seizures in the boy [14,22]. The available data show that the ketogenic diet has the properties of reducing (to a smaller or greater extent) epileptic seizures, frequently by several score percentage points or 50–90% of patients; a 90% reduction in seizures may be experienced by about 27% of patients. In some patients, complete remission was observed [94,95,96,97,98,99]. In 2022, an extensive study was conducted on 160 pediatric patients (mean age five years and nine months) with epilepsy treated with the ketogenic diet, and the effects were monitored for 3, 6, 12 and 24 months after its institution. An absence of seizures was observed (depending on the duration) in 13.7% of children after three months, 12.5% of children after six months, 14.4% after 12 months and 10.6% after 24 months. On the other hand, a reduction of ≥50% was observed in 41.9% of children after three months, 37.5% after six months, 28.7% after 12 months and 16.2% after 24 months [100]. A meta-analysis of 2022, based on the studies conducted on epileptic children, demonstrated a reduction in the number of seizures by at least 50%, together with a complete absence of seizures in as many as 48.31% of children. It was demonstrated that children on a ketogenic diet had a 5.6 times greater chance for seizure frequency reduction by at least 50% compared with a control group [101]. Another extensive study revealed that in patients observing a ketogenic diet for epilepsy, a reduction in seizure frequency by ≥50% occurred in 35–56.1% of the participants compared with 6–18% in the control group [102]. In 2020, a meta-analysis focused on the effect of the ketogenic diet on babies (<2 years) with epilepsy, based on a total number of 534 subjects. It was demonstrated that in as many as 33% of cases, a complete regression of seizures occurred, while in 59% of the babies, a significant reduction of seizure episodes by ≥50% was achieved. The authors have concluded that based on the results obtained, it can be said that the ketogenic diet is effective and safe in babies with drug-resistant epilepsy [103]. Currently, instead of the classic ketogenic diet, increasingly frequently, the modified Atkins diet (MAD) is used, which also is a model of the ketogenic diet. In one meta-analysis, the two nutritional models were compared in respect of better effectiveness. In the group of patients on the classic ketogenic diet, the reductions in seizure frequency by ≥50% after 1, 3, 6, 12 and 24 months were (as a percentage of the patients) 62%, 60%, 52%, 42% and 46%, respectively, while after 1, 3, 6 and 12 months in the patients using the modified Atkins diet, the percentages were 55%, 47%, 42%, and 29%, respectively. In view of that, the authors concluded that the effectiveness of both models was similar [104]. The authors of another meta-analysis drew a similar conclusion. They mentioned, however, a slight advantage of the modified Atkins diet in respect of the lower number of potential adverse events. Furthermore, the whole effect can also be indirectly influenced by the fact that MAD is, as a rule, tastier, thus encouraging better compliance with is requirements [93]. Two other meta-analyses also demonstrated similar results [102,105]. The ketogenic diet is also frequently prescribed in a version enriched with medium-chain triglycerides (MCTs) in view of their high ketogenicity. This was checked in one randomised controlled study, and it was revealed that the therapeutic effects were similarly favourable in both versions of the diet. The mean percentage of initial seizures was similar in the third, sixth and twelfth months. In the classic version of the diet, in the third month, it reached 66.5%; in the 6th month—48.5% and in the 12th month—40.8%. In the MCT-enriched version, it was 68.9% in the third month, 67.6% in the sixth month and 53.2% in the twelfth month [106]. The possible mechanisms of action of the ketogenic diet and the reduction in the number of epileptic seizures, in spite of over a hundred years of studies in this respect, are still not fully elucidated. Some potential mechanisms are, however, known and have an influence on the reduction of seizures; it is possible that they contribute to the observed therapeutic effect through synergistic actions [14,107]. The first of them is the possible anticonvulsant effect of the ketone bodies themselves, which was demonstrated in some studies [108,109,110,111,112] but not confirmed in other studies [113,114]. A study of 2019 has shed new light on that issue, presenting new molecular foundations of the anticonvulsant action of ketone bodies. It turns out that the anticonvulsant effectiveness of the ketogenic diet is positively correlated with the serum concentration of β-hydroxybutyrate, which has an ability to directly activate the potassium voltage-gated channel subfamily Q (KCNQ1/3) channels [115]. Another potential mechanism of ketone bodies’ action concerns the effect on neuronal metabolism (including mitochondrial function) and synaptic function. That mechanism postulates that it is glucose, which is readily available for neurons, the diffusion of which through the blood–brain barrier is quite easy (since the transport takes place in the endothelium of cerebral capillaries), that is indispensable for the initiation of neuronal seizure activity [116]. The ketogenic diet is supposed to reduce seizure frequency by decreasing the availability of glucose and, hence, energy and, in particular, the rate of its consumption by neurons. That has been confirmed in studies demonstrating a direct effect of glucose on the seizure threshold [117,118]. Some effects can be exerted by medium-chain triglycerides (MCTs), which are particularly frequently and willingly used in ketogenic diets. Apart from the properties of increasing the concentration of ketones in the body, they have the ability to increase decanoic acid concentration in plasma. The acid directly inhibits the α-amino-3-hydroxy-5-methyl-4-isoxazolepropionic acid receptor (AMPA) and, owing to that, shows an even better anticonvulsant effect than ketones [81,119,120]. In a publication in 2020, the extraordinary effectiveness of medium-chain fatty acids was confirmed since the addition of MCT oil to diets twice daily for three months was sufficient to reduce the number of seizures by 42% in adult patients with incurable epilepsy [121]. In the state of ketosis, an increase in mitochondrial metabolism is also observed, leading to an increase in ATP production, which activates ATP-sensitive potassium channels (KATPs), reducing, in turn, neuronal excitability [81,119,120]. It has also been demonstrated that one of the ketone bodies, that is, acetoacetate (Acac), inhibits the voltage-dependent Ca^2+^ channels (VDCCs) and decreases excitatory postsynaptic currents (EPSCs) at sites showing epileptic activity and, thus, inhibits convulsions in vivo [122]. Another potential mechanism is supposed to be the effect of ketones increasing the synthesis of the neurotransmitter GABA (through a reduction in aspartate concentration), the higher concentration of which can exert an inhibitory effect on convulsion initiation, and the synthesis of the neurotransmitter A1 adenosine, which can also show some anticonvulsant activity, reducing as well the level of excitatory glutamate [123,124,125,126]. Other possible mechanisms include the modulation of intestinal microbiota [127], the reduction of proinflammatory cytokine levels [128], an epigenetic effect of β-hydroxybutyrate through the inhibition of class I histone deacetylases and an effect on the transcription of genes (particularly those associated with the widely understood antioxidant factors) [129,130]. The most recent publications also discuss the above-mentioned mechanisms and suggest that, most likely, a synergy of all of them enables the antiepileptic effect of the ketogenic diet [131].

## 4. The Role of the Ketogenic Diet in the Therapy of Alzheimer’s Disease (AD)

Alzheimer’s disease (AD) is the most frequent cause of dementia worldwide. It is estimated that 50–70% of dementia cases are caused by this disease. The problem of dementia is very serious since it is estimated that by 2050, its incidence will triple. Currently, about 10 million new cases are noted annually. Taking into account the approximate current number of over 55 million people affected by the disease worldwide, its tripled incidence in a not-too-distant future seems to foreshadow a real calamity [132,133]. Alzheimer’s disease (AD) develops for many years, frequently not showing earlier any specific symptoms. Therefore, the use of adequate early diagnostic procedures and the institution of therapeutic management are frequently difficult. Early symptoms in the form of minor memory disturbances are frequently not regarded as an onset of Alzheimer’s disease (AD). With time, as a result of progressing brain neurodegenerative processes, the disturbances exacerbate, starting a new stage of disease advancement. Among other symptoms, problems occur with performing everyday activities, memorizing the meaning of words, disorientation and disorders of mood and sleep. As a result, the patient affected by the disease is frequently unable to function unassisted due to the occurrence of neurological and psychiatric symptoms [134,135,136]. The mortality rate due to Alzheimer’s disease (AD) has increased between 2000 and 2018 by as much as 146.2%. The disease has become the fifth most frequent cause of death among elderly people in America, which illustrates the scale of the problem [137]. In the brain of patients with Alzheimer’s disease (AD), increased amounts are found of β-amyloid (βA) and hyperphosphorylated tau protein (tau-p), which aggregate to form intracellular neurofibrillary tangles [138]. The factors contributing to Alzheimer’s disease (AD) development are multifaceted. They include, among other factors: depression and long-term stress, diabetes mellitus, hypertension, dyslipidaemia, obesity, cardiovascular diseases, traumatic cerebral injury, hyperhomocysteinaemia, oral cavity diseases, loss of hearing, sleep disorders, low physical activity, tobacco smoking, alcohol consumption, vitamin D deficiencies, inadequate diet, air pollution, poor education level and avoidance of social contacts [135]. 

The ketogenic diet can exert a favourable effect on Alzheimer’s disease (AD) in many mechanisms [139]. First, it is worth mentioning that the disease is frequently called type 3 diabetes mellitus, which involves the brain. This results from the fact that it shows molecular and biochemical features that are observed in type 1 and type 2 diabetes. Moreover, diabetic patients are at a significantly higher risk of AD development [140,141]. An important argument to support this is the fact of brain insulin resistance occurrence in patients with the disease. Insulin resistance developing in the brain of AD patients and disturbed signalling processes are well-grounded in the literature [141,142,143]. Socio-economic progress has contributed to people increasingly choosing processed products that belong to the western diet pattern. This, undoubtedly, is the cause of the development of not only diabetes mellitus but also, as demonstrated in a study in 2022, of AD, leading to brain insulin resistance and the progression of neurodegenerative processes [144]. In the course of AD, a reduction is observed in the amounts of GLUT1 and GLUT3, the two main glucose transporters in the brain. This is correlated with tau protein hyperphosphorylation and the density of neurofibrillary tangles in the brain, i.e., the typical signs of AD [145]. Insulin resistance and a reduced amount of glucose transporters in the brain cause neurons to have significantly hindered access to energy sources, so brain functioning disturbances are not surprising. A then-instituted, adequately composed and customised ketogenic diet can, in such cases, exert its effect on at least two main domains. On one hand, an induction of the ketosis state and an increase in ketone bodies concentration would provide an alternative source of energy for the brain (ketone bodies). In view of that, energy generation from glucose (problematic in the brain of AD patients) will not be indispensable anymore for the normal work of the brain since it can function excellently using ketone bodies, the transport of which (contrary to glucose) into the brain is not impaired in AD patients [146,147]. On the other hand, the ketogenic diet (particularly in combination with calorie deficits) would act as a causal treatment. It shows the ability to reduce insulin and glucose concentrations, and thus, insulin resistance, which is the cause of brain function disorders in AD, will be consistently reduced. The ketogenic diet can reduce insulin resistance through a number of mechanisms [52,148,149]. The most common method of determination of insulin resistance is the calculation of the Homeostatic Model Assessment–Insulin Resistance (HOMA-IR) index. Many studies have demonstrated a significant reduction of the index value due to the application of the ketogenic diet, which is a direct confirmation of its effect on insulin resistance. It was demonstrated, among other findings, that after 12 weeks, the HOMA-IR index value was reduced by 62.5% (from the initial 3.73 value to 1.4) [150]. In another study, just four weeks were enough to reduce the index by almost half (45.9%) [151]. The effect of the ketogenic diet on disturbed brain bioenergetics in AD and possible mechanisms of reduction of amyloid plaques has been increasingly suggested [152]. Animal studies have already demonstrated that, compared with a standard diet, the ketogenic diet results in the reduced deposition of amyloid plaques in the hippocampus, the decreased activation of the microglia and the improvement of cognitive functions, including learning and spatial memory [67]. In another study, a 25% reduction in amyloid plaques was found in mice with AD after the application of the diet [58]. The ketogenic diet can also act by exerting an effect on the expression of genes associated with neurodegenerative diseases such as AD, including genes associated with the metabolism of the hippocampus, and it prevents disorders of oxidative phosphorylation [153,154,155]. In AD, the function of mitochondria is also abnormal, leading to a gradual loss of their ability to produce energy. That phenomenon is related to inflammatory processes and the accumulation of amyloid plaques. It was demonstrated, however, that a ketogenic diet is able to induce the formation of new mitochondria through the activation of mitogenesis-regulating pathways. It also reduces the inflammatory condition in the brain, resulting from, among other factors, the excessive production of reactive oxygen species (ROS) by dysfunctional mitochondria [156,157,158,159]. Evidence shows that diet supplementation with medium-chain fatty acids from MCTs, similar to this case, seems to be extremely helpful, and this has been mentioned in an increasing number of publications [36]. The fatty acids show high ketogenicity, and the body is able to transform them in a simple process into ketones. A study in 2018 in patients with mild and moderate AD demonstrated that supplementation with a 30 g daily dose of MCTs contributed to the doubling of the uptake of ketones in the brain and increasing the brain’s total energy metabolism. Ketones produced from MCTs compensated for glucose deficiency in the brains of AD patients, proportional to the concentration of ketones in plasma [160]. MCT oil can maintain or improve cognitive functions in AD patients in a significant majority of cases, at about 80% [161]. It has also been demonstrated that supplementation with MCTs contributed to, among other results, an improvement of the working memory and cognitive functions in patients with AD as well as in individuals without dementia [162,163]. In an animal model, the effect of MCTs was also demonstrated on mitochondrial function improvement and the alleviation of the unfavourable action of β-amyloid on cortical neurons and the reduction of its total amount [58,164,165,166,167]. Moreover, the use of MCT oil seems helpful since it facilitates the maintenance of the high ketogenicity of the diet, even with a slightly increased consumption of carbohydrates (in relation to the ketogenic diet without MCT oil) [168]. A double-blind, placebo-controlled study also demonstrated that increased serum β-hydroxybutyrate concentrations contributed to an improvement in cognitive functions and memory [169]. In the case of AD, supplementation with exogenous β-hydroxybutyrate for 20 months also produced an improvement in cognitive functions, mood and everyday functioning [170]. The first randomised controlled study in patients with an unequivocal diagnosis of AD, assessing the effect of the ketogenic diet on the disease, was published in 2021. It compared the effect of the diet with that of a standard diet based on low-fat content. Compared with the low-fat diet, in the group of patients on the ketogenic diet, an improvement in cognitive function by 2.12 ± 8.70 points on the Addenbrooke’s Cognitive Examination III (ACE-III) scale, an improvement in everyday functioning by 3.12 ± 5.01 points on the Alzheimer’s Disease Cooperative Study Activities of Daily Living Inventory (ADCS-ADL) scale and an improvement in the quality of life by 3.37 ± 6.86 points on the Quality of Life in AD (QOL-AD) scale were demonstrated. It was also noted that the adverse effects were mild, while changes in cardiovascular risk parameters were mostly favourable [171]. Importantly, in spite of frequent accusations of problems with compliance with ketogenic diet requirements, half of the patients decided to continue it after the 12 weeks of study duration. The significant change in the quality of life of AD patients may be even more pronounced than the effect of drugs, including cholinesterase inhibitors, which exert an inconsistent influence on the quality of life [172,173]. Taking these results into account, the low-fat recommendations in AD should definitely be verified and challenged with the current results of scientific research.

## 5. The Role of the Ketogenic Diet in the Therapy of Parkinson’s Disease (PD)

Parkinson’s disease (PD) is a frequently observed neurodegenerative disease of the brain, the incidence of which has doubled since 1990. It develops particularly in elderly people, namely, in 1% of individuals aged over 60 years, although it is increasingly diagnosed in persons at a young age. It is a significant cause of disability worldwide since in 2019, it was responsible for 5.8 million disability-adjusted life years (increased by 81% in relation to 2000). At least 53 million people worldwide struggle with that disease, and in 2019, it was the cause of 329 thousand deaths, which was an increase of over 100% in relation to the year 2000. The disease is manifested with motor sluggishness, tremor, equilibrium disturbances and dysaesthesia or neuropsychiatric signs. The cause of these signs is damage to the neurons in the substantia nigra responsible, inter alia for dopamine production [1,174,175,176,177,178,179]. In this case, the ketogenic diet also may prove effective, and that effect is more and more frequently the subject of studies.

The ketogenic diet can affect Parkinson’s disease (PD) through several mechanisms resulting from the nature of the disease. Although specific unequivocal causes have still not been established, a persistent inflammatory condition of the nervous system, mitochondrial dysfunction, reactive oxygen species (ROS) excess, a reduced ability to produce dopamine, abnormal cerebral glucose metabolism and the accumulation of damaged proteins, so-called Lewy bodies composed of misfolded α-synuclein, have been observed [180,181]. It has been found that the ketogenic diet can affect each of the aspects mentioned. The anti-inflammatory effect in all neurodegenerative diseases has been described earlier, and in this respect, the diet has a multifaceted activity [182]. However, studies are available on the anti-inflammatory effect of the diet strictly in PD. Among other studies, a publication in 2022 demonstrated that the anti-inflammatory effect of the ketogenic diet in the disease is related to a modulation of the Akt/GSK-3β/CREB signalling pathway, mediated by the acetylation of metabotropic glutamate receptor 5 (mGluR5) promoter region histones in a rat Parkinson’s disease model [183]. This study confirmed mainly the neuroprotective effect of preventive ketosis compared to receiving KD as a therapeutic diet in the lipopolysaccharide (LPS)-induced rat PD model. After the induction of PD (with LPS), the model showed an increased regulation of proinflammatory mediators (TNF-α, IL-1 and IL-6), the loss of dopaminergic neurons, a reduction in mGluR5+ microglial cells, an increase in TSPO+ microglial cells, a reduction in H3K9 acetylation in the mGluR5 promoter region, and mGluR5 mRNA reduction with a decrease in the phosphorylation levels of the Akt/GSK-3/CREB pathway. These disturbances were improved by the dietary intervention of preventive KD in particular. PET imaging enabled the noninvasive detection and monitoring of the anti-inflammatory effect on PD (via the KD diet) related to histone acetylation or the DNA methylation of the mGluR5 gene. 

KD suppressed the inflammatory response (neuroinflammation) relevant to microglial activation and had a neuroprotective influence. The anti-inflammatory effect of KD on PD was related to the modulation of the mGluR5/Akt/GSK-3β/CREB signalling pathway by increasing the level of histone acetylation of the mGluR5 promoter region.

In addition, the pathological processes of neuroinflammation connected with PD are supposed to ameliorate by the multiple neuroprotective mechanisms of KD-induced ketosis involving the inhibition of proinflammatory mediator gene expression, the inhibition of the NLRP3 inflammasome assembly, epigenetic adaptations associated with calorie restriction, polyunsaturated fatty acids, ROS reduction, and the gut microbiome. Ketone bodies serve not only as an energy substrate but also as a signalling molecule. Finally, microglial cells can be modulated by various epigenetic mechanisms (DNA methylation and histone acetylation) and, thus, regulate neuroinflammation, resulting in neuroprotection [183].

Damaged mitochondria, ROS excess and abnormal glucose metabolism are closely interrelated, and the ketogenic diet can have an influence on those as well. Ketone bodies show an ability to rebuild new mitochondria to increase mitochondrial respiration and the production of ATP molecules. The reduction of free radicals thus also results from the improved efficiency of the respiratory chain in mitochondria [45,81,184]. The main ketone body, β-hydroxybutyrate, can also reduce the dopaminergic neurodegeneration and mitochondrial deficit observed in PD [38,185]. The neuroprotection of dopaminergic neurons seems to be of great importance, taking into account the scale of the problem of dopamine deficiency. Levodopa (L-DOPA), i.e., a dopamine precursor, has been, for years, one of the main drugs prescribed for PD. That drug can, however, contribute to the increased aggregation of α-synuclein, which, abnormally tangled, causes the formation of the Lewy bodies present in PD [186]. It has been found, however, that when using levodopa together with a ketogenic diet, the results can be far more favourable than those of the treatment with the drug alone (through an improvement of its bioavailability) [187,188,189]. Studies on PD animal models have clearly demonstrated that ketone bodies reduce dopaminergic neuronal death (BHB administered in vitro to cortical neurons and subcutaneously infused in mice) [38,190], decrease the number of proinflammatory cells in the brain and improve motor functions [62]. Another potentially possible mechanism of the ketogenic diet’s influence on PD is the indirect effect mediated by changes in the intestinal microbiome. It is known that diet significantly influences intestinal microbiome remodelling; on the other hand, it is known that the microbiome plays a great role in the pathogenesis and course of PD [191]. In 2018, a randomised controlled study was conducted on 47 patients with PD (38 of whom completed the study), who were divided into two groups: a group in which the ketogenic diet was applied and a group on a low-fat and high-carbohydrate diet. Although, after eight weeks, an improvement was observed in both groups, it was more pronounced in the group of patients on the ketogenic diet. In Part I of the Unified Parkinson’s Disease Rating Scale (UPDRS), the results in the ketogenic diet group improved by 41% in relation to the initial values (−4.58 ± 2.17 points), while those in the low-fat group improved by 11% (−0.99 ± 3.63 points). That was true of non-motor signs, while, in respect of motor signs, both groups presented a similar significant improvement. The adverse effects observed in both groups included increased sensations of hunger in the low-fat group and periodical exacerbations of tremor and/or stiffness in the ketogenic diet group [192]. A study in 2022 conducted on 16 patients with PD observing a ketogenic diet for 12 weeks also demonstrated its favourable effects. A significant improvement was noted on the Parkinson Anxiety Scale (PAS). At the same time, an improvement was observed in body mass, BMI, waist circumference, the concentrations of C-reactive protein (CRP), glycated haemoglobin, fasting insulin, HDL cholesterol and triglycerides. On the Center for Epidemiologic Studies Depression Scale (CESD-R-20), no major changes in depression symptoms were found [193]. In another study, among other findings, it was demonstrated that in PD patients, ketogenic diet application resulted in a mean reduction in Unified Parkinson Disease Rating Scale (UPDRS) result values by 10.72 points (a reduction by almost half compared with the initial results). An improvement was observed in respect of posture equilibrium, gait, resting tremor, mood and energy level, and it was achieved in just 28 days [188]. Another study comparing the ketogenic diet with a high-carbohydrate diet in PD patients demonstrated that in spite of the absence of motor function differences in the UPDRS between the groups, in the group of patients on the ketogenic diet, better results were observed in respect of short-term memory and verbal fluency [194]. In 2022, a case report was published on a 69-year-old woman with PD and mild depression and anxiety symptoms, to whom the ketogenic diet had been applied. A reduction in depression symptoms by 8 points on the Center for Epidemiologic Studies Depression Scale (CESDR) (from 42 to 34) and an improvement in the Parkinson Anxiety Scale (PAS) by 6 points (from 23 to 17) were noted. A significant improvement occurred in all health biomarkers, including a reduction in cardiovascular disease risk. On the other hand, in the UPDRS, an increase was observed from 24 to 33 points [195]. An improvement in UPDRS also occurred in all five study participants with PD, strictly complying with ketogenic diet requirements (fats 90%, proteins 8%, carbohydrates 2%) for 28 days [194]. Włodarek also referred to these results in his publication in 2019 [36]. Another study assessed the effect of the ketogenic diet on the quality of voice, which is decreased in PD patients. A Voice Handicap Index (VHI) test was used, and an improvement was observed in voice quality parameters in patients with PD, suggesting a possible alternative therapy for the improvement of that parameter in PD patients [196]. Taking into account the wide range of the effect on many aspects and potential therapeutic possibilities of the ketogenic diet in PD, further studies in this respect seem extremely necessary.

## 6. The Role of the Ketogenic Diet in the Therapy of Multiple Sclerosis (MS)

Multiple sclerosis (MS) is a neurodegenerative, inflammatory disease of the central nervous system (affecting the brain and spinal cord) of autoimmune origin. It concerns about 2.8 million people of either sex worldwide, including young individuals. It is also the main cause of disability among young people. Its incidence is unfortunately increasing; in 2013, it affected 2.3 million patients. It consists of damage to the myelin sheaths protecting neurons, thus causing disorders in the transmission of nerve impulses. Its manifestations take various forms, not infrequently different in different individuals (it can take a progressive form as well as a relapsing–remitting form). The frequently present symptoms include tingling sensations, limb weakening, problems with equilibrium, fatigue, dizziness, vision disorders and dysaesthesia [197,198,199,200].

Currently, a body of objective evidence that suggests a potentially favourable effect of the ketogenic diet in the treatment of MS is available. It can affect the course and prophylaxis of the disease while simultaneously offering safety of use and feasibility [201]. Taking into account the demyelination processes observed in MS, an effect of the ketogenic diet is suggested for the possible reconstruction and repair of the myelin sheaths. A possible even greater influence on these processes seems to be exerted by the ketogenic diet in the Mediterranean model, which has been suggested by the authors of a publication in 2022 [202]. The ketogenic diet affects the concentration of the brain-derived neurotrophic factor (BDNF), which is the main neurotrophic growth factor produced by neurons participating in myelin repair. The diet acts through a ketone body, i.e., β-hydroxybutyrate, which penetrates the blood–brain barrier, and through its effect on mitochondrial respiration and NF-KB, indirectly increasing BDNF synthesis through the activation of p300/EP300 histone acetyltransferase. Moreover, an inverse relationship has been demonstrated between serum glucose concentration and the amount of BDNF [203,204,205,206]. The Mediterranean model of the diet could exert an even greater effect in view of the increased amount of polyphenols in such a diet. It was shown that they activate the nuclear CREB factor and, thus, can additionally increase the amount of BDNF [207]. Another study also concerned the Mediterranean model of the ketogenic diet in 26 patients with MS. After four months on the diet, a significant intensification of the sensation of satiety (with similar values of ghrelin) and an increase in lean body mass and paraoxonase 1 (PON1) levels were observed. The authors explicitly suggest the favourable effect of the Mediterranean (isocaloric) ketogenic diet on the metabolism of their patients and associate the increase in the sensation of satiety with the reduction in inflammatory conditions and oxidation processes based on the observed changes of the studied parameters [208]. Another mechanism of action of the diet in MS therapy is its effect on the serum neurofilament light chain (sNfL), which is associated with multiple sclerosis (MS) and can serve as a marker of that disease. This was observed in a study on patients with relapsing–remitting MS. It was noted that the diet, six months after its institution, decreased sNfL levels, thus showing a neuroprotective effect in MS [209]. An extensive anti-inflammatory effect was also observed, specifically in patients with the disease. This was demonstrated in a six-month-long randomised controlled study of 60 patients. Compared with the control group, in the ketogenic diet group, a significantly reduced expression was noted of the arachidonate 5-lipoxygenase (ALOX5) gene, which encodes the enzymes for the biosynthesis of proinflammatory eicosanoids. Compared, however, to the results of the same individuals before and after the institution of dietary therapy, a significantly impaired expression was noted of other proinflammatory enzymes, i.e., cyclooxygenase 1 (COX-1) and cyclooxygenase 2 (COX-2), and an inverse correlation was observed between the expression of proinflammatory genes and patients’ quality of life in the Multiple Sclerosis Quality of Life-54 (MSQOL-54) scale [210]. In a study in 2022, among 65 participants with recurrent MS, in whom the ketogenic diet was applied for six months, some promising results were observed. Benefits were noted, among other findings, in the form of improvement in neurological disability, quality of life, depression or inflammatory conditions. A reduction occurred, by almost half, in the fatigue and depression results reported by the study participants. An improvement occurred not only in mental health but also in physical health. In the participants, an improvement was also seen in the mean values of the disability status in the Expanded Disability Status Scale (EDSS), from 2.3 ± 0.9 to 1.9 ± 1.1 points [211]. That has also been confirmed by an earlier study in 19 patients on the ketogenic diet for three months and 16 patients using it for six months. An improvement occurred in the results of depression and fatigue, body mass decreased, and the level of serologic proinflammatory adipokines was reduced, with good tolerance and safety of the diet [212]. No significant clinical improvement was, however, observed in another study on MS patients using ketogenic diets enriched with MCTs, but an evident reduction of fasting glucose and insulin levels was noted [213]. Taking into account the nature of the ketogenic diet, mimicking fasting, the possible additional mechanisms of its action are known and have been demonstrated in another randomised controlled study. It has been shown that the fast-mimicking diet increases regeneration in the oligodendrocyte precursors and remyelination in axons and reduces the symptoms of autoimmunisation and, thus, the symptoms of MS. Moreover, it is able to reduce the concentrations of proinflammatory cytokines, T helper type 1 cells (TH1), T helper type 17 cells (TH17) and antigen-presenting cells (APCs). The study has also provided evidence of potential benefits resulting from the ketogenic diet in the treatment of patients with relapsing–remitting MS (RRMS) [214]. Taking into account the multifaceted favourable effect of the ketogenic diet in MS, further studies could effectively lead to a change in the therapeutic approach to the disease.

## 7. Ketogenic Diet in the Therapy of Migraine

Migraine is the most frequently occurring neurological disease. It occurs in 12% of the world population, and chronic migraine (CM) is observed in 1–2% of people worldwide. Among the individuals struggling with episodic migraine, 2.5% of patients develop its chronic form. It is manifested with frequent paroxysmal headaches, which are frequently very intense. It is, therefore, a disease that significantly impairs patients’ quality of life and willingness to function on a daily basis. Individuals with migraine are at a higher risk of the development of other symptoms, including psychic disorders, sleep disorders or cardiovascular manifestations [215].

The ketogenic diet in the case of migraine shows significant potential benefits, which have been observed in an increasing number of recent clinical studies. Although the causes of migraine have not been unequivocally established, disorders in mitochondrial functioning and related problems with ATP production are supposed to belong to the possible mechanisms [216]. In such cases, the cause may be a disturbed metabolism of the cerebral cells, for which the possibility of taking energy from glucose is reduced; thus, ketone bodies seem to be a promising option here since they provide an alternative form of energy to the brain. However, looking not at the theory but at the actual results of studies in patients suffering from migraine, interesting conclusions can be drawn. A randomised controlled study comparing the effect of a very-low-calorie ketogenic diet with that of a very-low-calorie non-ketogenic diet demonstrated an advantage of the ketogenic diet. The study was conducted on 35 patients with migraine and overweight. The participants on the ketogenic diet, compared with the non-ketogenic diet, demonstrated a mean reduction of −3.73 migraine days monthly and −3.02 migraine attacks monthly. The percentage of patients with at least a 50% reduction in the mean number of days with migraine in the ketogenic group was 74.28%, compared with only 8.57% in the non-ketogenic group. It was suggested that diet can be a useful therapeutic strategy for migraine [217]. Another study in 2022, conducted on 23 patients with migraine and overweight, revealed a reduction in the number of days with headaches from 12.5 ± 9.5 days monthly on average, before the institution of the diet, to 6.7 ± 8.6 days on average after its institution. Furthermore, a reduction in the number of days taking drugs on an ad hoc basis was observed—from 11.06 ± 9.37 days monthly on average, before the institution of the ketogenic diet, to 4.93 ± 7.99 days after the institution. Moreover, body mass, fatty tissue mass and, thus, BMI values were also reduced in these patients. The favourable influence goes beyond the effect of body weight loss, and the authors concluded that other mechanisms must underlie the therapeutic activity [218]. The aim of another study in 2022 was an assessment of the effect of the ketogenic diet on treatment-refractory migraine in 22 patients (in the first study) and 31 subjects (in the second study). Very importantly, the ketogenic diet was compared with a low-carbohydrate diet. This means that the effects observed resulted strictly from the status of ketosis and not from the fact of the minimisation of carbohydrate contents in the diet. In the first group of 22 patients on the ketogenic diet, a significant reduction was observed in the frequency of migraine attacks, headache intensity and the amount of drugs taken. For comparison, in the low-carbohydrate diet group, no major changes were noted. In the study of the second group of 31 patients, similar effects were observed, which was evidence of the effectiveness of the ketogenic diet in the case of treatment-resistant migraine. A relationship was also observed between the production of ketones and their effect on headaches [219]. Another study also analysed the effect of the nutritional model on drug-resistant migraine. Patients with drug-resistant migraine were subjected to dietary intervention in the form of a ketogenic diet lasting three months. Very promising results were obtained since the mean number of days with pain decreased from 30 to 7.5 and the mean duration of pain episodes decreased to 5.5 h. Initially, 83% of the study participants reported maximal pain levels (on a pain rating scale). After the institution of the ketogenic diet, 55% of the study participants observed a reduction in pain intensity [220]. In 2022, for the first time, a study was conducted in order to systematise the data concerning the effect of ketogenic diets on migraine. The authors concluded that in most studies analysed, ketogenic diets reduced the number of attacks and their intensity in patients suffering from migraine [221].

## 8. Limitations and Potential Adverse Health Impacts of the Ketogenic Diet

The main limitations include a low number of available studies and, particularly, a low number of qualitative studies. Although the highest-quality body of evidence suggests the effectiveness of the ketogenic diet in the treatment of epilepsy, the situation is not similar in the case of the remaining neurological diseases. This results from the fact that the development of research in the field of neurological diseases other than epilepsy is a relatively new phenomenon. All meta-analyses and systematic reviews concerning Alzheimer’s disease, Parkinson’s disease, multiple sclerosis and migraine begin in 2020 (according to the PubMed search engine). This fact justifies the need for the exploration of this new research domain. 

Potential adverse effects of the use of the ketogenic diet are possible, but they rarely occur and usually result from inadequately adjusted diets. It should be kept in mind that the therapeutic approach to patients should be maximally individualised. Such effects were described in a meta-analysis in 2020, involving a total number of 932 participants [93]. It demonstrated that in epileptic children on ketogenic diets, gastrointestinal signs occurred most frequently. Among other effects, diarrhoea, constipation and vomiting were observed. The signs also occurred in children treated by other methods; therefore, they could not be ascribed strictly to the ketogenic diet. The remaining adverse effects, less frequently described in the meta-analysis, included body mass loss, infections, nausea, dysphagia, and lethargy. In adults, on the other hand, among other symptoms and signs, the following were rarely observed: headache, abdominal pain, irregular menstruation, drowsiness, and nephrolithiasis. In the study by Arora et al., no side effects of the ketogenic diet were found in adults with traumatic brain injury. The diet proved safe [43]. In a randomised controlled study on patients with Alzheimer’s disease, no major adverse effects were observed [171]. In a study conducted on patients with Parkinson’s disease, some of them presented periodical exacerbations of tremor and/or stiffness, increased irritability, and excessive hunger or thirst [192]. It should be kept in mind, however, that the period of the so-called “keto flu” is frequently regarded as an adverse effect. That period is, in fact, a transient episode frequently occurring at the beginning of treatment with the diet, and it poses no threat [222]. 

## 9. Summary

Taking into account all the data discussed, the ketogenic diet is undoubtedly a very promising model in the therapy of the above-mentioned neurological diseases. More than a hundred years of studies on the ketogenic diet’s effect on neurological diseases (starting with epilepsy) means that they belong to the main fields of research related to the therapeutic potential of the diet. This results from its very wide, pleiotropic effect on the body as well as from a number of (including those not yet known) mechanisms of action on the nervous system. Its favourable activity in neurological diseases, demonstrated in clinical studies, is related to the following: reducing the production of reactive oxygen species (ROS); reducing neuronal inflammatory conditions; the reconstruction of neuronal myelin sheaths; the repair of damaged mitochondria and the formation of new mitochondria and, thus, the effect on the disturbed neuronal metabolism in a number of neurological diseases; the provision of an alternative energy source for neurons in the form of ketone bodies; a reduction in glucose and insulin concentrations; the induction of autophagy; the reduction of microglia stimulation; the reduction of the excitatory postsynaptic current (EPSC) through action on voltage-dependent Ca^2+^ channels (VDCC); intestinal microbiota modulation and gene expression (epigenetic origin); assistance in the production of indispensable dopamine; and an increase in glutamine conversion into the neurotransmitter GABA. Together, with all the mentioned mechanisms, it is not surprising that the ketogenic diet in clinical studies shows a favourable effect on a number of neurological diseases, including epilepsy, Alzheimer’s disease (AD), Parkinson’s disease (PD), multiple sclerosis (MS) and migraine, which has been demonstrated in this paper. This is shown in Table 1. At present, a large body of indirect evidence suggests the effectiveness of the diet, while in the last few years, studies have increasingly aimed at demonstrating its actual direct effect on neurological patients. The currently available scientific data suggest a promising influence of the nutritional model in the therapy of neurological diseases. Despite that, there is a definite need to continue the further development of research in this field. Keeping in mind the exponentially growing incidence of neurological diseases, this approach could contribute in the future to a potential application of the diet in the clinical therapy of neurological patients, thus increasing the quality of life and lifespan of millions of people worldwide.

## Figures and Tables

**Figure 1 nutrients-14-05003-f001:**
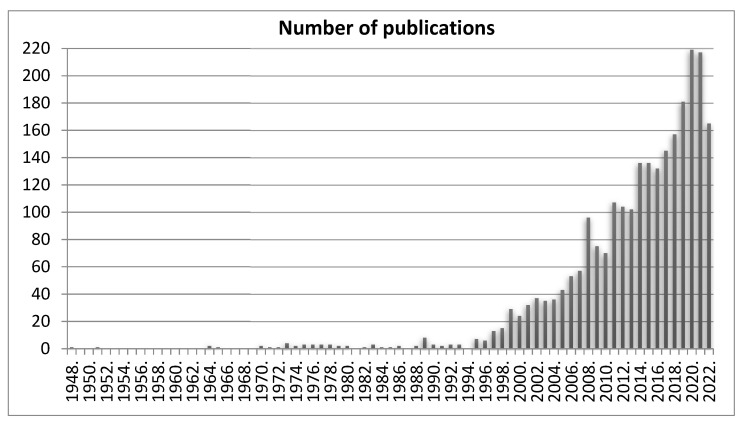
The number of publications for the entry “ketogenic diet neurological disease” in the PubMed base in the period from 1 January 1948 to 17 November 2022. Date of search: 17 November 2022.

**Figure 2 nutrients-14-05003-f002:**
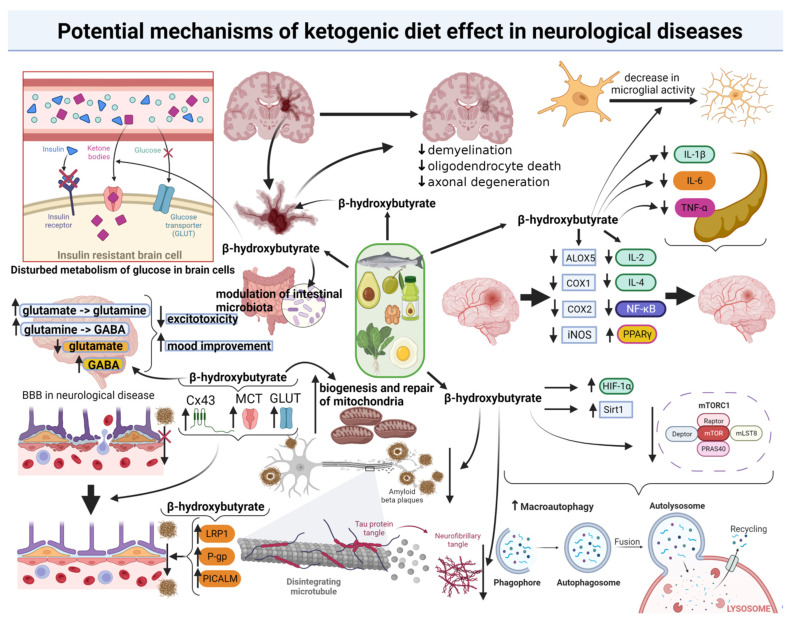
Potential mechanisms of the ketogenic diet effect in neurological diseases. GABA: Gamma-Aminobutyric Acid; BBB: blood–brain barrier; Cx43: connexin-43; MCT: monocarboxylate transporters; GLUT: glucose transporters; LRP1: LDL receptor-related protein 1; P-gp: glycoprotein P; PICALM: phosphatidylinositol binding clathrin assembly protein; IL-1β: interleukin-1β; IL-6: interleukin-6; TNF-α: tumor necrosis factor α; ALOX5: arachidonate 5-lipoxygenase gene; COX1: cyclooxygenase 1; COX2: cyclooxygenase 2; iNOS: inducible nitric oxide synthase; IL-2: interleukin-2; IL-4: interleukin 4; NF-κB: nuclear factor kappa-light-chain-enhancer of activated B cells; PPARγ: peroxisome proliferator-activated receptor γ; HIF-1α: hypoxia-induced factor 1α; Sirt1: sirtuin 1; mTORC1: mammalian target of rapamycin complex 1; mLST8: mammalian lethal with SEC13 protein 8; PRAS40: proline-rich Akt substrate of 40 kDa; mTOR: mammalian target of rapamycin. The above figure was created with BioRender.com, accessed on 23 November 2022. Agreement number: TF24OJVLEL.

**Table 1 nutrients-14-05003-t001:** Review of selected randomised controlled studies concerning the effect of the ketogenic diet on the discussed neurological diseases.

Disease	No. of Patients	Intervention in the Study Group	Intervention in the Control Group	Result	Reference
Alzheimer’s disease	26	12-week ketogenic diet; 58% fat (including 26% saturated and 32% unsaturated), 29% protein, 7% roughage and 6% carbohydrates net. + a multivitamin preparation *	12-week diet according to New Zealand’s principles of healthy nutrition; 11% fat (3% saturated, 8% unsaturated), 19% protein, 8% roughage, and 62% carbohydrates net. + a multivitamin preparation *	Compared with the control group, the ketogenic diet improved the following: cognitive functions (by 2.12 pts on the ACE-III scale), everyday functioning (by 3.13 pts on the ADCS-ADL scale), quality of life (by 3.37 pts on the QOL-AD scale)	[171]
Parkinson’s disease	47	8-week ketogenic diet, 1750 kcal, 152 g fat (67 g saturated), 75 g protein, 16 g carbohydrates net and 11 g roughage + a possibility of additional “loading” of 500 kcal (50 g fat (22 g saturated), 6 g protein, 5 g carbohydrates net and 4 g roughage)	8-week low-fat diet 1750 kcal, 42 g fat (10 g saturated), 75 g protein, 246 g carbohydrates net and 33 g roughage + a possibility of additional “loading” of 500 kcal (4 g fat (1 g saturated), 6 g protein, 102 g carbohydrates net and 11 g roughage)	Improvement in Part I of the UPDRS by 48% compared with 11% improvement in the control group, a greater improvement of non-motor signs in the ketogenic group, improvement of motor signs in both groups	[192]
Multiple sclerosis	60	6-month ketogenic diet, >160 g fat, ≤100 g protein, <50 g carbohydrates net, high (unspecified) roughage consumption	6-month diet according to the principles of a healthy diet in the German population	Reduced expression of proinflammatory ALOX5 compared with the control group, impaired expression also of other proinflammatory enzymes (COX1, COX2), significant inverse correlation between expression of proinflammatory ALOX5 and COX2 and the MSQoL-54 marker	[211]
Migraine	35	4-week very-low-calorie ketogenic diet (VLCKD), 20 g fat, ≥75 g protein, 30–50 g carbohydrates + a preparation with microelements	4-week very-low-calorie non-ketogenic diet (VLCnKD), 20 g fat, ≅50 g protein, ≥70 g carbohydrates + a preparation with microelements	Reduction in the mean number of days with migraine monthly by 3.73 days, compared with the control group, in the number of migraine attacks over that time by 3.02; a greater percentage of patients in the ketogenic group, in whom a reduction by at least 50% of the days with migraine, was obtained (74.28% vs. 8.57% of the participants)	[218]
Epilepsy	145	12-month ketogenic diet enriched with MCT, 70–75% energy from fat (30% long-chain fatty acids, 40–45% MCT), 10% energy from protein, 15% energy from carbohydrates	12-month classic ketogenic diet; in most cases, the fat to carbohydrates and protein ratio was 4:1; protein was maintained in amounts recommended by the WHO	Effectiveness of both types of ketogenic diets: a reduction of the mean number of initial attacks in the 3rd, 6th and 12th months in the classic version was 66.5%,48.5%, and 40,5%, respectively, while in the MCT version, it was 68.9%, 67.6%, and 53.2%, respectively	[106]

* Multivitamin and Mineral Boost, Clinicians Ltd., Auckland, New Zealand. ACE-III: Addenbrooke’s Cognitive Examination III; ADCS-ADL: Alzheimer’s Disease Cooperative Study Activities of Daily Living Inventory; QOL-AD: Quality of Life in AD; UPDRS: Unified Parkinson’s Disease Rating Scale; ALOX5: arachidonate 5-lipoxygenase; COX1: cyclooxygenase 1; COX2: cyclooxygenase 2; MSQOL-54: Multiple Sclerosis Quality of Life-54; MCT: medium-chain triglycerides; WHO: World Health Organization.

## Data Availability

Not applicable.

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
