# Peer review of "The Role of Ketogenic Diet in the Treatment of Neurological Diseases"

_nutrients, 2022, doi:10.3390/nu14235003_

Round 1

Reviewer 1 Report

The present review is aimed at illustrating the effect of ketogenic diet in neurological diseases. The paper is clear, well organized and well written. The topic is thoroughly treated, critically commented, and supported by a large number of updated references.

Major point:

The text probably lacks a paragraph on Alzheimer's disease since the citations from 131 to 174, all relating to Alzheimer's, are missing.

Minor points:

A figure should be added summarizing the possible cellular mechanisms that explain the effect of the ketogenic diet.

Line 413: ‘animals with PD’ should be ‘PD animal models’

Reference 190 is not appropriate

Author Response

Dear Reviewer 1.

With regards, Agnieszka Paziewska.

Reviewer 2 Report

The purpose of this review is to provide a comprehensive examination of the scientific data that supports the use of the ketogenic diet as a treatment for neurological disorders such as epilepsy, Alzheimer's disease, Parkinson's disease, multiple sclerosis, and migraines. The manuscript is interesting, but some issues must be addressed. Please, see my comments below:

Major

Please add Figures summarizing the molecular mechanisms of the ketogenic diet in the brain. Is there a possible link for all the diseases reviewed in this manuscript?

Some vital information about the manuscripts mentioned in the review is missing. For instance, the authors state that “Among other studies, a publication of 2022 demonstrated that the anti-inflammatory effect of the ketogenic diet in that disease was related to a modulation of the Akt/GSK-3β/CREB signaling pathway, mediated by acetylation of mGluR5 promoter region histones [183]”. This information is vague because the authors did not mention the model, region, or cell type involved in these findings. Another example sits on lines 413 to 416. Please check that along with the manuscript.

Lines 409 and 410 – the sentence “Therefore, it is not the best solution and is not a causal treatment, but it only enables minimisation of PD symptoms and signs is very vague and does not sound scientific.

Please add a section reviewing the limitations and possible adverse health impacts of the ketogenic diet.

Check abbreviations along the text. I indicated some below, but there are many more in the manuscript.

Minor

Lines 26 and 27: please check abbreviations

The information presented o lines 64-69 is very similar to the information provided on lines 96-98. This is a bit confusing. Please check.

Line 134: replace GNDF with BNDF

Line 196: please check the COX-2 abbreviation

Line 206: check Alzheimer’s disease abbreviation

Line 243: please check if the anatomical terminology “cerebral” is correct – “brain” would be more appropriate

Author Response

Dear Reviewer,

with regards, Agnieszka Paziewska

Round 2

Reviewer 1 Report

The authors improved their article in response to the comments made and it is now acceptable in the present form.